# Association between Pancreatoblastoma and Familial Adenomatous Polyposis: Review of the Literature with an Additional Case

**DOI:** 10.3390/genes15010044

**Published:** 2023-12-27

**Authors:** Andrea Remo, Silvia Negro, Riccardo Quoc Bao, Edoardo d’Angelo, Rita Alaggio, Gino Crivellari, Isabella Mammi, Rossana Intini, Francesca Bergamo, Matteo Fassan, Marco Agostini, Marco Vitellaro, Salvatore Pucciarelli, Emanuele Damiano Luca Urso

**Affiliations:** 1Pathology Unit, ULSS9 “Scaligera”, 37122 Verona, Italy; andrea.remo@aulss9.veneto.it; 2General Surgery 3, Department of Surgical, Oncological and Gastroenterological Sciences, University of Padova, 35121 Padua, Italy; quocriccardo.bao@unipd.com (R.Q.B.); m.agostini@unipd.it (M.A.); puc@unipd.it (S.P.); edl.urso@unipd.it (E.D.L.U.); 3Pathology Department, Ospedale Pediatrico Bambino Gesù, IRCCS, 00165 Roma, Italy; 4Familial Cancer Clinic and Oncoendocrinology, Veneto Institute of Oncology, IOV-IRCCS, 35121 Padua, Italy; gino.crivellari@iov.veneto.it (G.C.); isabella.mammi@iov.veneto.it (I.M.); 5Oncology 1, Veneto Institute of Oncology, IOV-IRCCS, 35121 Padua, Italy; rossana.intini@iov.veneto.it (R.I.); francesca.bergamo@iov.veneto.it (F.B.); 6Department of Medicine-DIMED, University of Padova, 35121 Padua, Italy; matteo.fassan@unipd.it; 7Unit of Hereditary Digestive Tract Tumors, Fondazione IRCCS Istituto Nazionale dei Tumori, 20133 Milan, Italy; marco.vitellaro@istitutotumori.mi.it

**Keywords:** pancreatoblastoma, familial adenomatous polyposis, inherited tumours syndromes

## Abstract

Background: Adult pancreatoblastoma (PBL) is a rare pancreatic malignancy, with recent evidence suggesting a possible link to familial adenomatous polyposis (FAP). This study aims to review the latest evidence and explore a possible association between adult PBL and FAP. Methods: Two independent literature reviews were conducted: (1) on PBL and FAP, and (2) on PBL in the adult population not diagnosed with FAP. Results: Out of 26 articles on PBL and FAP screened, 5 were selected for systematic review, including 1 additional case. We identified eight FAP-related PBL cases, with a median age of 40 (IQR: 34–50). Of these, seven (87%) occurred in adults. We found 65 cases of adult PBL not FAP-related; thus, 7 out of 65 cases (10.7%) of adult PBL reported in the literature are associated with a clinical diagnosis of FAP or were carriers of *APC* germline pathogenic variants (GPVs). Conclusion: Data suggest a non-random association between adult PBL and FAP. Further research is essential to optimise surveillance protocols and develop more effective treatment strategies.

## 1. Introduction

Familial adenomatous polyposis (FAP) is a rare hereditary syndrome characterised by the early onset of hundreds to thousands of colonic adenomas, with a lifetime risk of colorectal cancer of approximately 100%. This condition results from a germline mutation in the adenomatous polyposis coli (*APC*) gene, a tumour suppressor gene located on chromosome 5. Following a dominant inheritance pattern with almost complete penetrance, FAP is also associated with duodenal adenomas, desmoids and other neoplastic (e.g., thyroid and duodenal carcinoma, hepatoblastoma, medulloblastoma) and non-neoplastic features (e.g., congenital hypertrophy of the retinal pigment epithelium and supernumerary teeth) [1,2]. While the gastrointestinal manifestations of FAP have been extensively investigated, recent evidence suggests a possible association between FAP and a specific type of pancreatic malignancy known as pancreatoblastoma (PBL) [3,4]. PBL is a rare and aggressive form of pancreatic cancer, primarily affecting children and young adults. It accounts for approximately 25% of pancreatic neoplasms in children, while it remains an extremely rare entity in the adult population [5,6]. PBL exhibits aggressive behaviour, frequently associated with local invasion, recurrence and distal metastases. It typically appears as a circumscribed mass with cystic solid components, which poses a challenge in determining the presence of local infiltration [7,8,9,10]. Radical surgery is considered the best treatment option, and resection of metastases has also been contemplated [11,12,13]. Recent genetic research has revealed key molecular alterations in PBL, including the loss of heterozygosity of chromosome 11p, which is associated with Beckwith–Wiedemann syndrome in pediatric patients [14]. Furthermore, alterations in the Wnt/β-catenin pathway are widespread, with CTNNB1 or *APC* gene mutations being frequent [4,15]. To date, only about 70 cases of adult PBL have been reported in the literature, and due to its rarity and the limited number of reports, the pathogenesis of adult PBL remains largely unknown [3]. Currently, emerging evidence suggests a possible association between FAP and adult PBL. Several case reports and small studies have documented PBL in individuals with FAP, suggesting a possible genetic connection [3,4]. However, the exact mechanism by which the *APC* gene may lead to PBL still remains unclear. The aim of this study is to present the case of a patient diagnosed with FAP who subsequently developed adult PBL. In addition, we conducted a systematic review of the recent literature in search of a possible association between adult PBL and FAP to postulate a rare but non-random association between these two conditions.

## 2. Case Report

A 56-year-old woman with FAP was admitted to the University Hospital of Padua, complaining of dyspepsia and upper abdominal pain. She was a carrier of a deleterious transversion C to A at nucleotide 2805 of the *APC* gene, causing a premature stop codon and resulting in a truncated protein (Y935X). Her medical history revealed that she underwent a total colectomy with low ileorectal anastomosis at the age of 21 due to extensive colonic polyposis. Subsequent regular surveillance revealed no extracolonic manifestations except for gastric fundic gland polyps and duodenal adenomas. Some 35 years after colectomy, clinical examination suggested a mass in the left hypochondrium. This suspicion was confirmed by a CT scan, which showed a well-defined retroperitoneal mass measuring 19 × 8.5 cm. The mass originated from the tail of the pancreas with no evidence of nodal or distant metastases (Figure 1). The patient underwent a radical distal splenopancreatectomy. Histologically, the neoplasm showed acinar differentiation with squamous nests and strong, diffuse cytoplasmic and nuclear β-catenin staining (Figure 2). Systemic therapy was not recommended, and the patient remains alive with no evidence of local recurrence or distant metastases exceeding 100 months of follow-up.

## 3. Materials and Methods

Two independent literature searches were designed to identify: (1) studies that reported a histological diagnosis of PBL and a clinical and/or molecular diagnosis of FAP, and (2) studies that reported a histological diagnosis of adult PBL (i.e., age ≥ 18 years) without clinical and/or molecular diagnosis of FAP. A comprehensive search of the Pubmed, Ovid and Scopus archives was performed using the following medical subject heading (MeSH) search terms: (1) pancreatoblastoma, familial adenomatous polyposis, and (2) pancreatoblastoma, adult (Appendix A). Articles from 1946 to 31 September 2023 were screened. The reference lists of all articles found were also searched to identify additional relevant studies. Articles not written in English were excluded. We included articles that met our eligibility criteria, represented by PBL in FAP patients and adult PBL in those not diagnosed with FAP. Two reviewers (S.N and R.Q.B.) independently performed literature searches and matched their results according to the established inclusion criteria. Data on authors, year of publication, patient numbers, demographics, tumour characteristics, oncological outcomes and treatment modalities were analysed and recorded separately by the reviewers. A dedicated database of selected papers was created, and disagreements were resolved by two additional blinded reviewers (A.R. and E.D.L.U.). The systematic review was conducted according to the Preferred Reporting Items for Systematic Review and Meta-Analyses (PRISMA) (Figure 3, Appendix A) guidelines. This study was approved by the AOUPD Ethics Committee (protocol number 007401, 24 November 2022). Data were reported in accordance with the tenets of the Declaration of Helsinki, and the patient gave informed consent for the study.

## 4. Results

### 4.1. Study Inclusion and Characteristics of Included Studies


*PBL and FAP*


A total of 26 articles were initially screened, and 13 met the criteria for full-text review. After removing unavailable and not-relevant articles, five studies were selected for the systematic review (Figure 3). Of these five included studies, three were retrospective multicentre studies, one was a case report, and one was a letter to the editor.


*PBL in the Adult Population*


A total of 409 studies were initially screened, and 180 were considered for full-text review (excluding those focused on: review of pancreatic tumours with acinar differentiation, n = 15; pseudo-papillary tumours of the pancreas, n = 11; miscellanea, n = 19). After removing unavailable and not-relevant articles, 44 studies were selected for the systematic review (Appendix A).

### 4.2. Synthesis of Clinicopathological Data


*PBL and FAP*


Information on demographics, tumour characteristics, treatment, and outcomes of FAP-related PB are summarised in Table 1. The five studies and our case report included a total of eight patients with PBL and FAP colonic phenotype or a germline *APC* deleterious variant: seven were adults and one was an infant. The median age at diagnosis of FAP-related PBLs was 40 (IQR: 34–50) years, with a female/male ratio of 3/1(6/2 cases). PBLs had no preferred location in the gland (head, isthmus or tail). The median tumour size was five (IQR: 3.4–7.5) cm. At a median follow-up of 32 (IQR: 13–100) months, four PBL patients were alive and three had died of the disease (follow-up data were not available for one patient). The age at diagnosis of FAP and of the PBL was reported for only six cases: in four of them, the pancreatic tumour was detected more than 10 years after the diagnosis of FAP. In one case, with an *APC* variant C.5503A>G (p.R1835G), PBL was diagnosed without any signs of polyposis coli. Information on the *APC* mutations found was only available in four cases, so a genotype/phenotype correlation was not possible. Data on systemic treatments administered to PBL were too limited to be reported in detail.


*PBL in the Adult Population*


Systematic review of the literature found 65 cases of adult PB not diagnosed with FAP. The median age at diagnosis of PBL was 37 (IQR: 27–61) years, with a female/male ratio of 1/1.5 (26/39 cases). The median tumour size was 5.8 (IQR: 4.3–9) cm (Appendix A).

## 5. Discussion

In this review, we investigated the characteristics of a rare tumour in the context of a rare syndrome. We found eight cases (including our additional case) of FAP-related PBL. Of these, seven (87%) occurred in the adult population. In total, 65 cases of PBL in the adult population not diagnosed with FAP have been described in the literature. Taken together, these findings indicate that more than 7 out of 65 cases (10.7%) of adult PBL may be associated with a clinical diagnosis of FAP or were carriers of an *APC* GPVs, suggesting a non-random association between adult PBL and FAP. It is crucial to identify rare clinical conditions associated with inherited syndromes, as this awareness significantly enhances the possibility of recognising the disease within the population and optimising surveillance for carriers. Numerous examples exist where clinical conditions initially considered extremely rare in the context of an inherited disease eventually become integral components of the syndrome. Muir, E.G., and Torre, D. independently made the initial reports of sebaceous neoplasms and keratoacanthomas associated with visceral malignancies in 1967 [19,20]. In 1991, Cohen, P.R., et al. collected 120 cases of sebaceous neoplasms associated with abdominal cancers, formalising the definition of Muir–Torre syndrome 25 [21]. However, it was only recently that Muir–Torre syndrome was officially associated with one of the clinical conditions related to a deficiency in the germline mismatch repair (MMR) system, specifically Lynch syndrome [22]. Similar considerations could be made for hepatoblastoma (HB) and FAP. The first association between FAP and HB was reported in 1983 [23]. Subsequently, more than 50 cases were documented in 2001 and by 2022 over 100 patients were identified [24]. Surveillance for HB in infants from FAP families is now recommended 28 [25].

PBL is considered to be a rare paediatric cancer. However, if we consider only FAP patients, there is only one report of PBL in children with FAP, as mentioned in a paper focusing on medulloblastoma and *APC* mutation carriers [18]. Although PBL is much less common in adulthood, eight cases of adult PBL in FAP have been reported. Taken together, these data suggest a non-random association between FAP and PBL, particularly in the adult population.

Excluding the one paediatric case, the median age of PBL diagnosis in our series of FAP patients was 40 years, which is similar to that reported for all patients with adult PBL not selected for FAP (median 37 years; IQR: 27–61; *p* = 0.6). Recently, a review by Omiyale et al. reported a 27-month OS of 58% for adults PBL [3]. Among these patients, 42% died due to the disease, 4% died due to unrelated causes, 16% survived with the disease, and the remaining 38% demonstrated no evidence of the disease (NED). In our study, even considering the small cases enrolled, four (50%) of FAP-related PBLs remained alive and with NED after a median follow-up of 46 (IQR:18.5–90) months. A possible explanation for the good outcome observed in our series is that the female gender may lead to a better cancer outcome, as is already the case for several cancers [26]. In support of this, the gender distribution showed a higher proportion of females (3:1 female to male ratio) in FAP-related PBL compared to adult PBL not selected for FAP (1/1.5 female to male ratio). On the other hand, since the largest FAP-related PBLs (6, 7.5 and 19 cm) were alive after a long follow-up period (60, 143, 100 months, respectively), the presence of a germline *APC* deficiency may indicate less aggressive behaviour of the tumour in this population. These data are in line with previous reports by Machado and Suemitsu et al., who reported better survival outcomes in those PBLs with a somatic mutation on the *APC* gene compared to those that were wild-type for the mutation [27]. Typically, FAP is diagnosed at an average age of 20 years. Therefore, based on our case series, the diagnosis of PBL would be expected to occur later, at around 40 years of age on average. However, there are cases where the diagnosis of PBL precedes that of polyposis. For instance, Ikenoue et al. reported a case of PBL in a patient with an attenuated form of familial adenomatous polyposis (AFAP) before the identification of the syndrome itself [28]. Furthermore, Yamaguchi et al. suggest that PBL could be diagnosed prior to the syndrome in individuals carrying a deleterious *APC* variant, even without a personal or familial history of polyposis [17]. Several previous studies have found an association between FAP and pancreatic cancer. Giardiello et al. analysed FAP families in the Johns Hopkins registry and found a relative risk of pancreatic adenocarcinoma in these patients of 4.46 (95% CI: 1.2–11.4) [29]. Karstensen et al. recently confirmed previous findings, reporting a risk of approximately 6.45 (95% CI, 2.02–20.64) for pancreatic cancer among FAP patients [30]. Despite the fact that current guidelines do not include FAP as an inherited condition requiring pancreatic surveillance, these results suggest that the spectrum of pancreatic malignancies may be more relevant in FAP patients than previously assumed, thus warranting regular monitoring for individuals with FAP syndrome. Complex surveillance programs have already been established internationally for those at the highest risk of pancreatic cancer (i.e., *CDKN2A* GPVs carriers, *BRCA1* and *2* GPVs carriers). Periodic magnetic resonance imaging and endoscopic ultrasounds could reduce the risk of advanced cancer in these populations [31]. Given the low incidence of pancreatic carcinoma in FAP patients, no recommendation could be made. However, the likelihood of pancreatic tumours suggests the need for a prompt evaluation of any signs or symptoms that could be related to pancreatic disease. Moreover, as somatic molecular profiling is now increasingly used in oncology, the results of our review suggest that if a somatic *APC* pathogenic variant is detected on PBL, germline investigation would be indicated.

Several limitations of this study should be discussed.

This is the first comprehensive review regarding the association of FAP and PBL. As a result of the rarity of the disease, there are limited published series available. Furthermore, most published studies are made up of small sample sizes and case reports. Additionally, there is a lack of homogeneity in the definition of polyposis coli across many studies, and the majority of them do not include molecular classification of the polyposis. However, even without the possibility of statistically proving a non-random association between PBL and FAP, our review provides good data to support the hypothesis of a reasonable association between FAP and adult PBL. Therefore, FAP care providers may be encouraged to report on PBL cases, and PBL care providers may be encouraged to screen for polyposis coli and germline *APC* mutation. It is possible that PBL in FAP patients is more common than previously thought, thus necessitating the development of new surveillance strategies to manage cancer risk.

## 6. Conclusions

PBL in the context of FAP is a rare and complex condition that poses challenges for both surveillance and treatment. While the genetic link between *APC* mutation and PBL development is becoming clearer, much still needs to be understood about this unique conceivable association. Further research is essential to improve our knowledge of adult PBL, optimise surveillance protocols and develop more effective treatment strategies.

## Figures and Tables

**Figure 1 genes-15-00044-f001:**
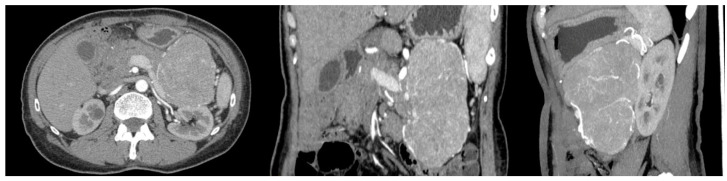
CT scan of the case report.

**Figure 2 genes-15-00044-f002:**
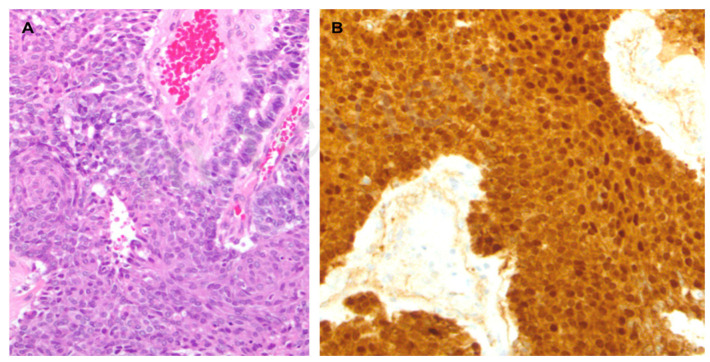
(**A**). Microscopic examination revealed a hypercellular neoplasm with a solid and acinar growth pattern admixed with pale/roundish areas with a whorled aspect, representing the so-called squamoid nestes (hematoxylin-eosin). (**B**). The neoplasm showed strong and diffuse cytoplasmic and nuclear β-catenin staining.

**Figure 3 genes-15-00044-f003:**
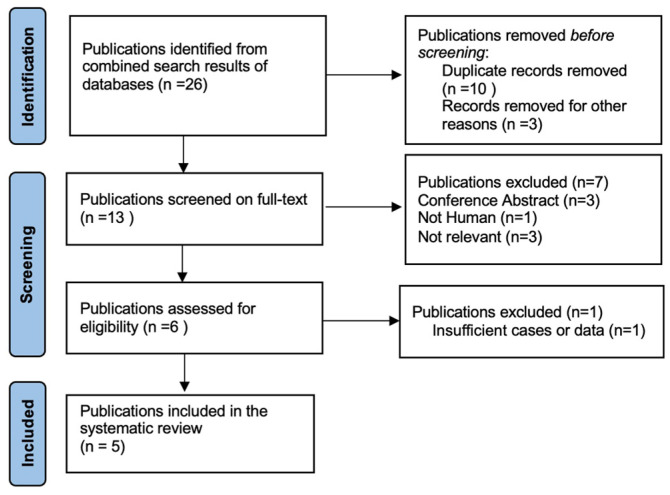
Selection of studies for review “PBL and FAP”.

**Table 1 genes-15-00044-t001:** Study participants’ characteristics “PBL and FAP”.

Study, y	Study Characteristics		Patient Characteristics					Tumourcharacteristics			Treatment	Outcome		
	Setting, Study Period	n	Age FAP Diagnosed, y	Sex, F/M	Germline APC GPVs	Age PBL Diagnosed, y	Lesion Associated, y	Location	Size, cm	Metastasis at Diagnosis		F.up, mo	Recurrence/ Distant Metastasis	Status
Abraham et al. 2001 [4]	Retrospective, Multicentre (1967–2000)	1	na	F	NM_000038.6(APC):c.3927_3931del (p.Glu1309Aspfs*4)	51	na	na	na	na	na	na	na	na
Moussata et al. 2015 [16]	Letter to the Editor (1990–2009)	1	25	M	na	35	Intestinal adenomas (25)	Isthmus	6	no	Surgery	60	no	NED
Yamaguchi et al. 2018 [17]	Case Report	1	37	F	NM_000038.6:c.5503A>G (p.Arg1835Gly)	37	No lesion associated	Head	5	no	Surgery + CT (Adriamycin, Gemcitabine, and Cisplatin)	13	yes	DOD
Reid et al. 2019 [15]	Retrospective, Multicentre (na)	3	na	F	na	50	Colorectal cancer (36), Maxillary osteoma (na), Intestinal adenomas (na), skin tumour (na)	Head	7.5	no	na	143	no	DOD
			na	F	na	43	Colorectal cancer (42)	Tail	3.4	no	na	14	no	NED
			na	M	na	34	na	Tail	2.5	yes	na	1	no	DOD
Massimino et al. 2021 [18]	Retrospective, Multicentre (2007–2016)	1	8	F	NM_000038.6(APC):c.3927_3931del (p.Glu1309Aspfs*4)	11	Medulloblas toma (na), thyroid cancer (na)	Head	3.5	no	Surgery	32	no	NED
Current Case	2021	1	25	F	NM_000038.6:c.2805C>A (p.Tyr935Ter)	56	Intestinal adenomas (35)	Tail	19	no	Surgery	100	no	NED

## Data Availability

No new data were created or analyzed in this study. Data sharing is not applicable to this article.

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
