# Peer review of "Association between Pancreatoblastoma and Familial Adenomatous Polyposis: Review of the Literature with an Additional Case"

_genes, 2023, doi:10.3390/genes15010044_

Round 1
Reviewer 1 Report
Comments and Suggestions for Authors
The authors present a unique case of an adult with pancreatoblastoma and FAP. The authors then conduct a comprehensive literature review to determine an association between adult-onset pancreatoblastoma and FAP.
The article is well-written. Some minor suggestions.
1. The mutations throughout the manuscript should be updated using HGVS nomenclature. Even older reports, should have the APC variant updated with reference to the appropriate NM transcript. Also an updated variant interpretation using ACMG guidelines should be conducted. Of interest there are updated Clinical Genome resource Guidance for APC (PMID 37800450).
2. there are numerous typos in table 1 please fix
3. The rationale for doing a literature search only for adult pancreatoblastoma should be more prominent. Why not look also at pediatric pancreatoblastoma?
4. the conclusion that there is an association with FAP and adult pancreatoblastoma may be pre-mature given there is no statistical analysis of this association with p-values.
Comments on the Quality of English Languageenglish is acceptable
Author Response
Reply to Reviewer 1):
The authors present a unique case of an adult with pancreatoblastoma and FAP. The authors then conduct a comprehensive literature review to determine an association between adult-onset pancreatoblastoma and FAP.
The article is well-written. Some minor suggestions.
Dear Reviewer, thank you for the time you spent and for your comments.
- The mutations throughout the manuscript should be updated using HGVS nomenclature. Even older reports, should have the APC variant updated with reference to the appropriate NM transcript. Also an updated variant interpretation using ACMG guidelines should be conducted. Of interest there are updated Clinical Genome resource Guidance for APC(PMID37800450).
I updated the mutations using the HGVS nomenclature. I have included them in the table 1.
- there are numerous typos in table 1 please fix
I have corrected typing errors and added the doi of the articles in Table 1.
- The rationale for doing a literature search only for adult pancreatoblastoma should be more prominent. Why not look also at pediatric pancreatoblastoma?
PBL is considered to be a rare paediatric cancer. However, if we consider only FAP patients, there is only one report of PBL in children with FAP, as mentioned in a paper focusing on medulloblastoma and APC mutation carriers [24]. Although PBL is much less common in adulthood, 8 cases of adult PBL in FAP have been reported. Taken together, these data suggest a non-random association between FAP and PBL, particularly in the adult population.
We included this annotation in the discussion section (Line 170)
- the conclusion that there is an association with FAP and adult pancreatoblastoma may be pre-mature given there is no statistical analysis of this association with p-values.
We changed one the last sentences of the discussion:
“However, the results of our review support evidence of an association between FAP and adult PBL”
With the sentence:”… However, even without the possibility to statistically prove a non-random association between PBL and FAP, our review provides good data to support the hypothesis of a reasonable association between FAP and adult PBL.” Line 224
In the conclusion we underlined the association of PBL and FAP in conceivable (and not demonstrated)
"While the genetic link between APC mutation and PBL development is becoming clearer, much still needs to be understood about this unique conceivable association." Line 234
Reviewer 2 Report
Comments and Suggestions for Authors
The review explores the association between familial adenomatous polyposis (FAP) and periampullary and duodenal adenocarcinomas (PBL). Among 146 rare syndromes, 8 FAP-related PBL cases (87% in adults) were identified. The findings suggest a non-random connection between adult PBL and FAP, However, the study acknowledges limitations, including the rarity of the disease, small sample sizes in published studies, and a lack of homogeneity in the definition of polyposis coli. Despite these limitations, the findings support evidence of an association between FAP and adult PBL, encouraging healthcare providers to report on PBL cases and consider screening for polyposis coli and germline APC mutation. The study is significant for enhancing understanding of the rare tumor PBL in the context of familial adenomatous polyposis (FAP), guiding targeted surveillance and personalized treatment approaches, correlating gender with better outcomes, encouraging reporting for comprehensive knowledge, and suggesting the need for new surveillance strategies in FAP patients.
Comments on the Quality of English LanguageOnly need minor revisions. For example:
Line 21: Instead of "7(87%) occurred in adults," use "Seven (87%) occurred in adults."
Line 23: expand the abbreviation "GPVs" for the first mention, and then use the abbreviation in subsequent mentions.
Author Response
Reply to Reviewer 2):
The review explores the association between familial adenomatous polyposis (FAP) and periampullary and duodenal adenocarcinomas (PBL). Among 146 rare syndromes, 8 FAP-related PBL cases (87% in adults) were identified. The findings suggest a non-random connection between adult PBL and FAP, However, the study acknowledges limitations, including the rarity of the disease, small sample sizes in published studies, and a lack of homogeneity in the definition of polyposis coli. Despite these limitations, the findings support evidence of an association between FAP and adult PBL, encouraging healthcare providers to report on PBL cases and consider screening for polyposis coli and germline APC mutation. The study is significant for enhancing understanding of the rare tumor PBL in the context of familial adenomatous polyposis (FAP), guiding targeted surveillance and personalized treatment approaches, correlating gender with better outcomes, encouraging reporting for comprehensive knowledge, and suggesting the need for new surveillance strategies in FAP patients.
Only need minor revisions. For example:
Line 21: Instead of "7(87%) occurred in adults," use "Seven (87%) occurred in adults."
Line 23: expand the abbreviation "GPVs" for the first mention, and then use the abbreviation in subsequent mentions.
Dear Reviewer 2, thank you for the time you spent and for your comments. I have corrected in the text all the suggestions you gave me.
Round 2
Reviewer 1 Report
Comments and Suggestions for Authors
The authors have sufficiently addressed my comments.